# Dynamic Range Reduction via Branch-and-Bound

## Abstract

A key strategy to enhance specialized hardware accelerators, such as GPUs and FPGA, is to reduce the numerical precision in arithmetic operations, which increases processing speed and lowers latency—both crucial for real-time AI applications. In this work, we consider NP-hard Quadratic Unconstrained Binary Optimization (QUBO) problems, which arise in machine learning and beyond. We show that these problems often require high numerical precision, posing challenges for hardware solvers. We introduce a principled Branch-and-Bound algorithm for reducing the precision requirements of QUBO problems by utilizing the dynamic range as a measure of complexity. Experiments demonstrate that our method reduces the dynamic range in subset sum, clustering, and vector quantization problems, thereby increasing their solvability on actual quantum annealers and FPGA-based digital annealers.

## 1. Introduction

Hardware acceleration is a major driving force in the recent advent of artificial intelligence (AI). Virtually all large-scale AI models rely on hardware accelerated training via Tensor Processing Units (TPUs), Graphics Processing Units (GPUs), or Field-Programmable Gate Arrays (FPGAs). A key ingredient of these accelerators is parallelism—a large computation is split into smaller pieces, solved via multiple compute units. Clearly, each compute unit must read its inputs from memory. However, memory bandwidth is limited. Hence, to achieve a large level of parallelism, the input that each compute unit needs must be as small as possible. To this end, model parameters with limited precision, e.g., 16-bit, 8-bit, or even smaller, are considered and special training procedures are employed to directly train models with low-precision parameters (Choukroun et al., 2019). AI

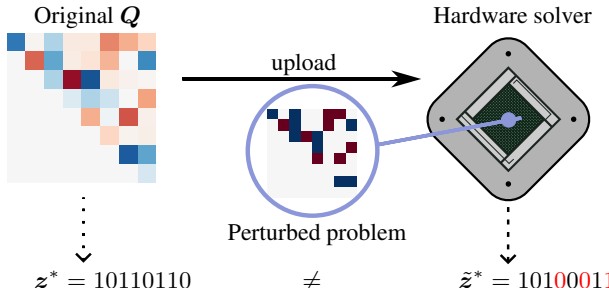

Figure 1: Illustration of parameter perturbation for finite precision hardware. The original QUBO $Q$ is perturbed through quantization errors which can lead to spurious optima.

accelerators usually rely on parallel implementations of basic linear algebra routines. There is, however, a multitude of AI problems whose inherent computational complexity does not stem from linear algebra operations. Examples include clustering (Aloise et al., 2009), probabilistic inference (Ermon et al., 2013), or feature selection (Brown, 2009). At the core of these tasks lies a combinatorial optimization (CO) problem. As of today, solving such ML-related CO problems exactly is out of reach for high-dimensional instances. However, analog devices (Yamamoto et al., 2020; Mohseni et al., 2022; Mastiyage Don et al., 2023), ASICs (Matsubara et al., 2020), FPGAs (Mücke et al., 2019; Kagawa et al., 2020), and quantum computers (Albash & Lidar, 2018) have recently made promising progress when it comes to solving CO problems. In particular, we consider Quadratic Unconstrained Binary Optimization (QUBO) (Punnen, 2022) and equivalent Ising problems

$$\min_{z \in \{0,1\}^n} z^\top Q z \Leftrightarrow \min_{s \in \{-1,1\}^n} s^\top J s + h^\top s , \quad (1)$$

where $Q$, $J$ and $h$ are real (proper definitions follow in Sec. 3). Despite (1)'s simple structure, it is NP-hard, and hence covers a plethora of real-world optimization challenges, from problems like the traveling salesperson and graph coloring (Lucas, 2014) to machine learning (ML) (Bauckhage et al., 2018; Mücke et al., 2023) and various other applications (Biesner et al., 2022; Chai et al., 2023). One common issue of QUBO hardware solvers, also called Ising machines, is limited physical precision of the matrix entries, as real-world hardware devices use finite numerical representations. It turns out that simply truncating decimal

[1]Anonymous Institution, Anonymous City, Anonymous Region, Anonymous Country. Correspondence to: Anonymous Author <anon.email@domain.com>.

Preliminary work. Under review by the International Conference on Machine Learning (ICML). Do not distribute.

digits of $\boldsymbol{Q}$ is not sufficient, since the resulting optimization problem will have different local and global optima (Mücke et al., 2025) (see Fig. 1). In the paper at hand, we develop an algorithmic machinery for reducing the numeric precision required to represent QUBO instances. The proposed method relies on the dynamic range (DR) of any QUBO matrix as a measure of its complexity.

We present a novel iterative procedure, which reduces the DR while preserving the optimal solution of the underlying QUBO. Moreover, we explain why greedy methods are likely to produce sub-optimal results and propose a novel method that is based on policy rollout (Bertsekas et al., 1997; Silver et al., 2016) to adress the greedy nature of baseline methods. Our contributions can be summarized as follows:

- We formulate the problem of reducing the required precision via a Markov decision process and introduce a fully principled Branch-and-Bound algorithm for exactly solving the problem in a finite number of steps.

- We propose effective and efficiently computable bounds for pruning the search space.

- Combining our algorithm with the well-known rollout policy, we further improve efficiency and performance.

- We support our theoretical insights with an experimental analysis for ML-related problems on quantum hardware and an FPGA-based digital annealer. The results demonstrate the effectiveness of our precision reduction, its superiority over baseline methods, and the benefits it provides for hardware solvers.

## 2. Related Work

It is well known that hardware devices tailored towards solving QUBO problems suffer from a limited parameter precision (Oku et al., 2020). While FPGA-based digital annealing devices (Şeker et al., 2022) have a fixed *bit-width* for representing problem parameters, e.g., 16-bit, quantum computers are prone to *integrated control errors* (Booth et al., 2017; Vyskočil et al., 2019). For preserving global optima, in (Oku et al., 2020) the bit-width is reduced by introducing exponentially many auxiliary variables dependent on the number of reduced bits. This can be combined with the heuristic rounding of the parameters (Yachi et al., 2022), which however, can lead to different optima. A similar method respecting the underlying hardware topology of a D-Wave quantum annealer can be found in (Mooney et al., 2019). Instead of enlargening the problem size, (Yachi et al., 2023) follows the approach of solving topologically equivalent instances, each with reduced precision requirements. The number of these instances is exponential in the number of reduced bits and optimum preservation is only guaranteed if all instances have the same global optimum.

A more general precision measure than the bit-width, which is only defined for integer parameters, is discussed in (Stollenwerk et al., 2019a;b). The *maximum coefficient ratio* of a QUBO problem is directly related to the performance on D-Wave quantum annealers. It is mentioned that it is largely affected by penalty parameters for incorporating constraints. (Verma & Lewis, 2022; Alessandroni et al., 2023) try to optimize these penalties by using bounds on the optima of the underlying problem. However, these methods are not applicable for arbitrary QUBO instances, e.g., when the large precision stems from the underlying data and not the problem formulation. In (Mücke et al., 2025), the *dynamic range* is identified as an improved complexity measure, and a method for iteratively reducing it is proposed. The method can be applied for any QUBO instance. The underlying idea is to update single QUBO matrix entries within specific interval boundaries, computed by bounding the optimal QUBO value. The method is guaranteed to preserve the original optima. However, the heuristics presented in that work are greedy and often get stuck in local optima. In what follows, we build upon (Mücke et al., 2025) by formulating a more elaborate algorithm that overcomes this issue.

## 3. Background

We denote matrices with bold capital letters (e.g. $\boldsymbol{A}$) and vectors with bold lowercase letters (e.g. $\boldsymbol{a}$). Sets will be symbolized by calligraphical or blackboard bold capital letters (e.g. $\mathcal{A}$, $\mathbb{A}$). Let the index set from 1 to $n$ be denoted as $[n] \coloneqq \{1, \ldots, n\}$. For an optimization problem, something optimal will be denoted by the superscript "*". We use " $\hat{\ }$ " or " $\check{\ }$ " to indicate a maximum/upper bound or a minimum/lower bound, respectively.

### 3.1. QUBO

A QUBO problem is completely characterized by an upper triangular matrix $\boldsymbol{Q} \in \mathbb{R}^{n \times n}$. The QUBO *energy* of a binary vector $\boldsymbol{z} \in \{0,1\}^n$ is defined as $E_{\boldsymbol{Q}}(\boldsymbol{z}) \coloneqq \boldsymbol{z}^\top \boldsymbol{Q} \boldsymbol{z} = \sum_{i \leq j} Q_{ij} z_i z_j$. Note that any real QUBO matrix can be brought into an equivalent upper triangular form. The objective of a QUBO problem is to find a binary vector $\boldsymbol{z}^* \in \{0,1\}^n$ which optimizes the QUBO energy

$$\boldsymbol{z}^* \in \mathcal{Z}^*(\boldsymbol{Q}) \coloneqq \underset{\boldsymbol{z} \in \{0,1\}^n}{\arg\min}\, E_{\boldsymbol{Q}}(\boldsymbol{z}) , \qquad (2)$$

where $\mathcal{Z}^*(\boldsymbol{Q})$ is the set of optimizers for the QUBO problem with matrix $\boldsymbol{Q}$. Let $\mathcal{Q}_n$ denote the set of upper triangular matrices in $\mathbb{R}^{n \times n}$. The problem in (1) is NP-hard (Pardalos & Jha, 1992), i.e., in the worst case, the best known algorithm is an exhaustive search over an exponentially large candidate space. Furthermore, any problem in NP can be reduced to QUBO with only polynomial overhead, making this formulation a very general form for combinatorial

optimization. A range of solution techniques has been developed over past decades, e.g., exact methods (Narendra & Fukunaga, 1977; Rehfeldt et al., 2023) with worst-case exponential running time, approximate techniques such as simulated annealing (Kirkpatrick et al., 1983), tabu search (Glover & Laguna, 1998), and genetic programming (Goldberg & Kuo, 1987); see (Kochenberger et al., 2014) for a comprehensive overview.

### 3.2. Dynamic Range

Although the entries of a QUBO matrix are theoretically real-valued, real-world computing devices have limits on the precision with which numbers can be represented. To quantify the precision required to accurately represent QUBO parameters, we adopt the concept of dynamic range (DR) from signal processing. For this, we first define the set absolute differences between all elements of $\mathcal{X} \subset \mathbb{R}$ as $\mathcal{D}(\mathcal{X}) := \{|x - y| : x, y \in \mathcal{X}, x \neq y\}$, and write $\check{D}(\mathcal{X}) := \min \mathcal{D}(\mathcal{X})$ and $\hat{D}(\mathcal{X}) := \max \mathcal{D}(\mathcal{X})$. For a given QUBO matrix $\boldsymbol{Q} \in \mathcal{Q}_n$ the DR is defined as

$$\mathsf{DR}(\boldsymbol{Q}) := \log_2 \left( \frac{\hat{D}(\mathcal{U}(\boldsymbol{Q}))}{\check{D}(\mathcal{U}(\boldsymbol{Q}))} \right) , \tag{3}$$

where $\mathcal{U}(\boldsymbol{Q}) := \{Q_{ij} : i, j \in [n]\}$. Note that always $0 \in \mathcal{U}(\boldsymbol{Q})$, since $\boldsymbol{Q}$ is upper triangular, that is $Q_{ij} = 0$ for $i > j$. A large DR indicates that many bits are required to represent all parameters of $\boldsymbol{Q}$ accurately in binary, as the parameters span a wide range of values and require fine gradations. Taking the next larger integer larger than the DR quantifies how many *bits* are required to faithfully implement the parameters of a QUBO matrix.

Different measures for describing the required precision are the *coefficient ratio* (CR) (Stollenwerk et al., 2019b)

$$\mathsf{CR}(\boldsymbol{Q}) = \frac{\max\{|Q_{ij}| : i, j \in [n]\}}{\min\{|Q_{ij}| : i, j \in [n]\} \setminus \{0\}} \leq 2^{\mathsf{DR}(\boldsymbol{Q})} ,$$

and the *bit-width* (BW) (Yachi et al., 2023)

$$\mathsf{BW}(\boldsymbol{Q}) = \lceil \log_2 \max\{|Q_{ij}| : i, j \in [n]\} \rceil + 1 \geq \mathsf{DR}(\boldsymbol{Q}) ,$$

which is only defined for integer entries. The BW does not capture inter-weight distances, making it an inaccurate measure when scaling parameters to a specific range. Even though CR might be very small, the DR can still be large, but the reverse does not hold. That is to say, we understand DR as an accurate measure of representational complexity.

## 4. Methodology

Our main goal is to reduce the DR of a given QUBO problem by transforming the corresponding QUBO matrix subject to the requirement that a global optimizer must be kept intact.

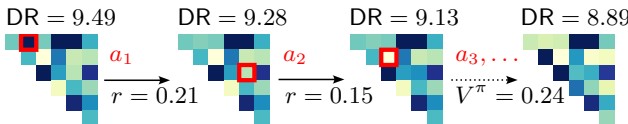

Figure 2: Illustration of the MDP described in Sec. 4. Every step $t$, we choose an action $a_t$ in form of an index pair and update our state $s_t$ to $s_{t+1}$ to obtain a matrix with a smaller DR. The goal is to maximize the value function $V^\pi$.

To this end, we define the notion of *optimum inclusion* ($\sqsubseteq_*$) on QUBO instances as $\boldsymbol{Q} \sqsubseteq_* \boldsymbol{Q}' \Leftrightarrow \mathcal{Z}^*(\boldsymbol{Q}) \subseteq \mathcal{Z}^*(\boldsymbol{Q}')$. We formulate the precision reduction problem as follows:

$$\underset{\boldsymbol{A} \in \mathcal{Q}_n}{\arg\min} \quad \mathsf{DR}(\boldsymbol{Q} + \boldsymbol{A}) \tag{4a}$$

$$\text{s.t.} \quad \boldsymbol{Q} + \boldsymbol{A} \sqsubseteq_* \boldsymbol{Q} . \tag{4b}$$

Let us give a small example for clarification.

**Example 1.** *Consider the following $2 \times 2$ matrices:*

$$\boldsymbol{Q} = \begin{bmatrix} 0.8 & -1.5 \\ 0 & -1000 \end{bmatrix}, \quad \boldsymbol{Q}' = \begin{bmatrix} 0.8 & -1.5 \\ 0 & -2 \end{bmatrix} .$$

*Observing the corresponding QUBO problems, we get*

$$\underset{\boldsymbol{z} \in \{0,1\}^2}{\arg\min} \boldsymbol{z}^\top \boldsymbol{Q} \boldsymbol{z} = \underset{\boldsymbol{z} \in \{0,1\}^2}{\arg\min} \boldsymbol{z}^\top \boldsymbol{Q}' \boldsymbol{z} = \begin{pmatrix} 1 \\ 1 \end{pmatrix} ,$$

*that is, $\boldsymbol{Q}$ and $\boldsymbol{Q}'$ have the same optimizer, therefore $\boldsymbol{Q} \sqsubseteq_* \boldsymbol{Q}'$. Furthermore, it holds that*

$$\boldsymbol{Q} + \boldsymbol{A} = \boldsymbol{Q}', \ \boldsymbol{A} := \begin{bmatrix} 0 & 0 \\ 0 & 998 \end{bmatrix} .$$

*When we compare the DR of $\boldsymbol{Q}$ and $\boldsymbol{Q}'$, we find that $\mathsf{DR}(\boldsymbol{Q}) \approx 10.29$, $\mathsf{DR}(\boldsymbol{Q}') \approx 2.49$.*

Example 1 demonstrates that, in principle, it is possible to reduce the DR while preserving an optimizer of the QUBO problem. Interestingly, we can find an optimal solution $\boldsymbol{A}^* = (\delta_{ij}(1 - 2z_i^*))_{i,j=1}^n - \boldsymbol{Q}$ to (4), when we already know an optimizer $\boldsymbol{z}^* \in \mathcal{Z}^*(\boldsymbol{Q})$. $\boldsymbol{Q} + \boldsymbol{A}^*$ is a diagonal matrix only consisting of the entries $-1, 0, 1$ with a minimum $\mathsf{DR}(\boldsymbol{Q} + \boldsymbol{A}^*) = 1$. Nevertheless, solving an arbitrary QUBO problem with matrix $\boldsymbol{Q}$ is NP-hard, but the resulting optimum of (4) is a diagonal matrix, for which the corresponding QUBO problem is solvable in linear time $\mathcal{O}(n)$. Taking the common assumption that $\mathsf{P} \neq \mathsf{NP}$, solving (4) is also NP-hard and thus as intractable to solve as the QUBO problem itself. Thus, we consider a slightly more constrained version than (4).

### 4.1. Markov Decision Process Formulation

We consider the state space as $\mathcal{S} = \mathcal{Q}_n$ and the action space $\mathcal{A} \subset [n] \times [n]$. The state transition function is given by

Figure 3: Exemplary depiction of the search space when applying our B&B algorithm (Sec. 5) to some QUBO matrix $\boldsymbol{Q}$. In every step, we expand our search space (from top to bottom, Sec. 5.1) and check whether a branch can be pruned ($r^* < \check{r}$, Sec. 5.3). The small filled circles indicate the visited states of our algorithm and the pruned parts are depicted as gray dashed lines. The fraction of pruned states is $40/85 = 0.47$ (without rollout). Since the search space size grows exponentially with the horizon $T = 5$, we execute a PR (Sec. 5.2) for $\tilde{T} = 2$ steps. From here on, a base policy is followed without expanding further, which is depicted in blue. The red path indicates the optimal solution and the yellow path shows the base policy.

$f(s, a) = f(\boldsymbol{Q}, (k, l)) = \boldsymbol{Q} + h(\boldsymbol{Q}, (k, l))\boldsymbol{e}_k\boldsymbol{e}_l^\top$, where $\boldsymbol{e}_k, \boldsymbol{e}_l$ are the standard basis vectors with zeros everywhere except at index $k$ and $l$, respectively. $h : \mathcal{Q}_n \times [n] \times [n] \to \mathbb{R}$, is a function for determining the parameter update and we consider the following heuristic (Mücke et al., 2025): Matrix entries are changed by a parameter $w$, $Q_{kl} \to Q_{kl} + w$ with $k, l \in [n], k \le l$ such that an optimizer is preserved, i.e., $y^-(\boldsymbol{Q}) \le w \le y^+(\boldsymbol{Q}) \Rightarrow \boldsymbol{Q} + w\boldsymbol{e}_k\boldsymbol{e}_l^\top \sqsubseteq \boldsymbol{Q}$. Details on heuristic $h$ and how to compute the bounds $y^-(\boldsymbol{Q})$ and $y^+(\boldsymbol{Q})$ can be found in (Mücke et al., 2025) and are given in the Appendix. Note that $y^-(\boldsymbol{Q}) \le 0 \le y^+(\boldsymbol{Q})$ and $h$ is illustrated in Fig. 4a. We define the reward as the change of DR, $r(s, a) := \mathrm{DR}(s) - \mathrm{DR}(f(s, a))$. The four-tuple $(\mathcal{S}, \mathcal{A}, f, r)$ defines a Markov decision process (MDP). Now assume that we want to change $T$ QUBO matrix entries such that the accumulated reward is maximized. Formally, the goal is to find a policy $\pi^* : \mathcal{S} \to \mathcal{A}$, s.t.,

$$\pi^* = \arg\max_{\pi:\mathcal{S}\to\mathcal{A}} V^\pi(s_0) = \arg\max_{\pi:\mathcal{S}\to\mathcal{A}} \sum_{t=0}^{T-1} r\left(s_t, \pi(s_t)\right) \quad (5a)$$

$$= \arg\min_{\pi:\mathcal{S}\to\mathcal{A}} \mathrm{DR}\left(f_T(\boldsymbol{Q}, \pi)\right) , \quad (5b)$$

where the equality follows through a telescopic sum and the $t$-time state transition $f_t$ following policy $\pi$ is defined as $f_t(s_0, \pi) := f\left(s_t, \pi(s_t)\right) = s_{t+1}$, $s_0 := \boldsymbol{Q}$. Our MDP is illustrated in Fig. 2. Observing that the transition is a simple matrix addition, we can write $f_T(\boldsymbol{Q}, \pi) = \boldsymbol{Q} + \boldsymbol{A}'$, where $\boldsymbol{Q} + \boldsymbol{A}' \sqsubseteq \boldsymbol{Q}$. Thus, the optimization objective of the decision process in (5) is a more restricted version of the problem in (4). That is, we do not optimize over the set of all optimum inclusive matrices, but over the subset of matrices which can be created with any policy following our MDP framework. The cumulative sum in (5a) is also called the value function $V^\pi$ for a policy $\pi$. Using the recursive *Bellman equation*

$$V^{\pi^*}(s_t) = \max_{a \in \mathcal{A}} \left[r(s_t, a) + V^{\pi^*}\left(f(s_t, a)\right)\right] , \quad (6)$$

we can find an optimal policy in (5) with dynamic programming (DP), using a shortest path-type method. In our case, we have no knowledge about the final state $f_T(\boldsymbol{Q}, \pi)$ and thus the search space is exponentially large (see Fig. 3). Choosing the number of iterations $T$ logarithmic in the problem dimension, i.e., $T = \log_2(n^2) = 2\log_2(n)$, results in a sub-exponential ($o(2^n)$) state space size. Thus, we have an asymptotically slower growth than the exponentially large state space size $2^n$ of the original QUBO problem. However, in practice, super-polynomial runtimes are often not tractable, especially for large $n$. Due to this fact, (6) is typically solved with approximate DP methods such as Monte Carlo Tree Search (MCTS), policy rollout (PR) or reinforcement learning (Bertsekas, 2019; 2021). We present a *Branch and Bound* (B&B) algorithm utilizing PR to reduce the complexity of solving (5).

## 5. Branch and Bound

Following all paths of possible QUBO matrix updates is intractable. We combine PR with the B&B paradigm to obtain a trade-off between computational complexity and solution quality—solution paths which cannot lead to an optimum are pruned, based on bounds on the best found solution. The algorithm is given in Fig. 3: the search space is expanded (branch) and every state is checked whether it can be pruned (bound). This is done until the final horizon $T$ is reached and the state with the minimum DR is returned.

### 5.1. Branch

In the *branch*-step, the search space is expanded from the current considered state. The question arises how to decide which indices to consider in the current iteration, i.e., which entries of QUBO matrix should be changed. The obvious method is to use all $n(n + 1)/2$ upper triangular indices of the whole matrix, which we will further indicate by ALL.

With large $n$, this expansion gets very large and thus we also consider a different method in our experiments. This method is based on the observation, that only four entries of the QUBO matrix affect the DR when changing a single weight. Namely the smallest/largest weight and the weights which are closest to each other, which will be denoted by IMPACT. This drastically reduces the search space search size and the performance to ALL is compared in Sec. 6.

### 5.2. Policy Rollout

Having an index pair $(k, l)$ at hand, a new state is created from the current state $\boldsymbol{Q}$ by following the transition function $f$. Instead of solving the problem in (5) exactly and expanding the search space for $T$ steps, we also consider a PR approach. It describes the concept of following a given a base policy $\dot{\pi}$ for a number of steps. For a given rollout depth $\tilde{T} \leq T$, we denote the policy which optimizes its path for $\tilde{T}$ steps and then follows $\dot{\pi}$ for $T - \tilde{T}$ steps as $\bar{\pi}_{\tilde{T}}$. We use a greedy policy $\dot{\pi}(\boldsymbol{Q}) := \arg\min_{a \in \mathcal{A}} \mathrm{DR}\left(f(\boldsymbol{Q}, a)\right)$ which myopically optimizes the DR when taking a single step. Since the solution quality is monotonically increasing with $\tilde{T}$, we obtain a trade-off between the size of the state space and the performance of our algorithm. Using PR is motivated by the well known *rollout selection policy*. At time $t$, the optimal future reward $V^{\pi^*}\left(f(s_t, a)\right)$ is approximated with the reward $V^{\dot{\pi}}\left(f(s_t, a)\right)$ following $\dot{\pi}$. The Bellman equation (6) is modified to

$$V^{\tilde{\pi}}(s_t) = \max_{a \in \mathcal{A}} \left[ r(s_t, a) + V^{\dot{\pi}}\left(f(s_t, a)\right) \right] . \quad (7)$$

The resulting rollout selection policy $\tilde{\pi}$ is at least equal and typically better than the base policy $\dot{\pi}$. It is renowned for its simplicity and strong performance, largely due to its close relationship with the fundamental dynamic programming algorithm of policy iteration. (7) represents the optimal one-step look-ahead policy, when subsequently following a base policy. This principle can be generalized to $\tilde{T}$-step look-ahead rollouts, $\tilde{T} \leq T$, where the solution quality increases with increasing rollout horizon $\tilde{T}$. The exact solution for (5) is obtained if $\tilde{T} = T$. Thus, PR can be seamlessly integrated into our B&B algorithm. An experimental comparison between $\bar{\pi}_{\tilde{T}}$ and $\tilde{\pi}$ with using the aforementioned variants can be found in Sec. 6.1.

### 5.3. Bound

We want to prune states, which cannot lead to the optimal solution. Deciding whether a state can be pruned, is dependent on bounds of the reachable best solution from that given state. Given the current best final DR $r^*$, we can prune a state $\boldsymbol{Q}$ if it is smaller than a lower bound $\check{r}(\boldsymbol{Q}, T)$ on the best reachable solution $\mathrm{DR}(f_T(\boldsymbol{Q}, \pi^*))$, i.e., if $r^* \leq \check{r}(\boldsymbol{Q}, T) \leq \mathrm{DR}(f_T(\boldsymbol{Q}, \pi^*))$. Pruning states, we do not have to expand the search further and can drastically

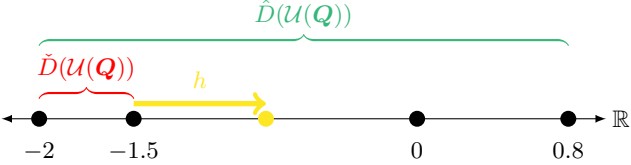

(a) We can read off $\hat{D}(\mathcal{U}(\boldsymbol{Q})) = 2.8$ (red) and $\check{D}(\mathcal{U}(\boldsymbol{Q})) = 0.5$ (green). Change of parameter $Q_{01}$ using a heuristic $h(\boldsymbol{Q}, 0, 1) = 0.7$ (yellow). The sorted parameters are given by $q_1 = -2$, $q_2 = -1.5$, $q_3 = 0$ and $q_4 = 0.8$.

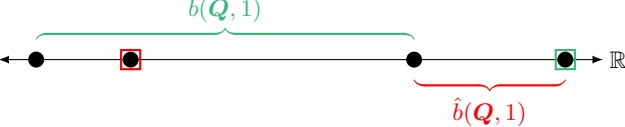

(b) Bounds when a single QUBO parameter is changed to 0: A lower bound (top, green) is given by $\hat{D}(\mathcal{U}(f_1(\boldsymbol{Q}, \pi^*))) \geq \check{b}(\boldsymbol{Q}, 1) = 2$ and an upper bound (bottom, red) by $\check{D}(\mathcal{U}(f_1(\boldsymbol{Q}, \pi^*))) \leq \hat{b}(\boldsymbol{Q}, 1) = 0.8$. The changed parameters are indicated with rectangular boxes.

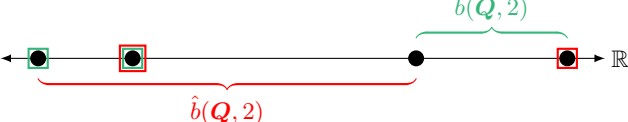

(c) Bounds when two QUBO parameters are changed, namely $\check{b}(\boldsymbol{Q}, 2) = 0.8$ and $\hat{b}(\boldsymbol{Q}, 2) = 2$. For details, see Fig. 4b.

Figure 4: Sorted QUBO matrix entries given in Example 1. Heuristic $h$ is depicted in Fig. 4a and the methods for finding a lower bound on the DR are given in Figs. 4b and 4c.

reduce the computation time. We find a lower bound on $\mathrm{DR}(f_T(\boldsymbol{Q}, \pi^*))$ with a lower/upper bound $\check{b}(\boldsymbol{Q}, T)/\hat{b}(\boldsymbol{Q}, T)$ on the numerator/denominator in (3)

$$\mathrm{DR}(f_T(\boldsymbol{Q}, \pi^*)) \geq \log_2\left( \frac{\check{b}(\boldsymbol{Q}, T)}{\hat{b}(\boldsymbol{Q}, T)} \right) =: \check{r}(\boldsymbol{Q}, T) .$$

Let $m := n^2$ be the number of entries of an $n \times n$ matrix. For any $\boldsymbol{Q} \in \mathcal{Q}_n$, there is an ordering (bijective map) $\sigma : [m] \to [n] \times [n]$ of entries such that $q_\ell \leq q_{\ell+1}$, $q_\ell \equiv Q_{\sigma(\ell)}$, $\forall \ell \in [m]$. With this notation, $\hat{D}(\mathcal{U}(\boldsymbol{Q})) = q_m - q_1$ and $\exists j \in [m-1] : \check{D}(\mathcal{U}(\boldsymbol{Q})) = q_{j+1} - q_j$. A visualization for an ordering of Example 1 is shown in Fig. 4a.

**Lower Bound on Maximum Distance** For finding a lower bound on $\mathrm{DR}(f_T(\boldsymbol{Q}, \pi^*))$, we optimistically assume that we can set all parameters to $0$ while maintaining an optimizer of $\boldsymbol{Q}$. Since $0$ is always considered in the computation of the DR, this corresponds to an optimal strategy of changing the parameters, because the DR cannot increase. Changing a single parameter, the numerator $\hat{D}(\mathcal{U}(\boldsymbol{Q}))$ in (3) is maximally reduced if we set $q_1/q_m$ larger/smaller than $q_2/q_{m-1}$. The maximum possible re-

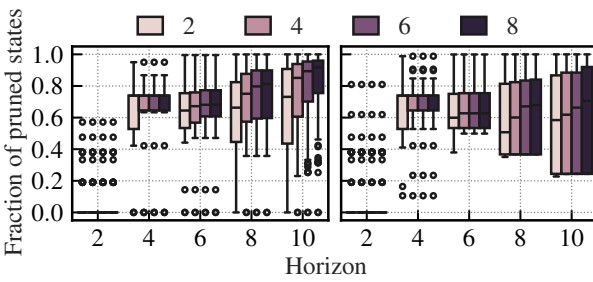

Figure 5: Fraction of pruned states. Different depths (2, 4, 6 and 8) for updating the current best DR are compared for $n = 8$ (left) and $n = 16$ (right).

duction is equal to $\min\{q_2 - q_1, q_m - q_{m-1}\}$. Iterating this process for $T$ times, we end up with $\check{b}(\boldsymbol{Q}, T) := \min\{q_{m-T+i} - q_{i+1} : 0 \leq i \leq T\}$. An illustration for Example 1 is found in Figs. 4b and 4c.

**Upper Bound on Minimum Distance** Obtaining a lower bound is a little more tricky and an iterative procedure is given in Algorithm 1. Since we are concerned with the smallest distance between two QUBO parameters, we consider the set of distances between "neighboring" parameters $\bar{\mathcal{D}}(\mathcal{U}(\boldsymbol{Q})) := \{q_{i+1} - q_i : i \in [m-1]\} = \{d_i : i \in [m-1]\}$. We iteratively set QUBO weights to 0, which are part of the minimum distance, maximizing the minimum distance. Define an ordering $\rho : [m-1] \to [m-1]$, s.t., $d_{\rho(i)} \leq d_{\rho(i+1)}$. It then holds that $\check{D}(\mathcal{U}(\boldsymbol{Q})) = d_{\rho(1)}$. $\check{D}(\mathcal{U}(\boldsymbol{Q}))$ is maximally reduced if we change $q_{\rho(1)+1}$ or $q_{\rho(1)}$, s.t., $d_{\rho(1)}$ is not the smallest distance anymore. We change the weight with the smaller corresponding distance (Lines 6 and 7, Algorithm 1). If $q_{\rho(1)}$ is changed, $d_{\rho(1)-1}$ is updated to $d_{\rho(1)-1} + d_{\rho(1)} = q_{\rho(1)+1} - q_{\rho(1)-1}$ and if $q_{\rho(1)+1}$ is changed, $d_{\rho(1)+1}$ is updated to $d_{\rho(1)+1} + d_{\rho(1)} = q_{\rho(1)+2} - q_{\rho(1)}$ (Algorithm 1, Algorithm 1). No update is required if $\rho(1) = 1$ or $\rho(1) + 1 = m$. The new smallest distance either equals the second smallest distance $d_{\rho(2)}$ or one of the two newly updated ones. The smallest distance $d_{\rho(1)}$ is removed (Line 12, Algorithm 1) from $\bar{\mathcal{D}}$ and the ordering $\rho$ is updated with the updated distances (Line 13, Algorithm 1). In Figs. 4b and 4c, this is illustrated for Example 1. The computational complexity of our bounds is derived and discussed in the Appendix.

## 6. Experiments

In what follows, we consider numerical experiments and study the impact of our method on a D-Wave Advantage System 5.4 quantum annealer (QA) and an FPGA-based digital annealer (DA) (Mücke et al., 2019). Three exemplary problems are considered: BINCLUS represents 2-means clustering, SUBSUM consists of finding a subset from a list of values that sum up to a given target value and VECQUANT

---

**Algorithm 1** LOWERBOUND

**Input:** $\boldsymbol{Q}, T$
**Output:** Lower bound $\check{r} \leq \mathsf{DR}(\mathcal{U}(\boldsymbol{Q}_T^{\pi^*}))$
1: $\check{b} \leftarrow \min\{q_{m-T+i} - q_{i+1} : 0 \leq i \leq T\}$
2: Compute $\sigma$, s.t., $q_\ell \leq q_{\ell+1}$, $q_\ell \equiv Q_{\sigma(\ell)}$   ▷ Sort weights
3: $\bar{\mathcal{D}} \leftarrow \{d_i : i \in [m-1]\}$, $d_i = q_{i+1} - q_i$
4: Compute $\rho$, s.t., $d_{\rho(\ell)} \leq d_{\rho(\ell+1)}$   ▷ Sort distances
5: **for** $t = 1$ to $T$ **do**
6:     $\mathcal{I} \leftarrow \{\rho(1), \rho(1) + 1\}$
7:     $i_* = \arg\min_{i \in \mathcal{I}} d_i$
8:     **if** $i_* = \rho(1)$ **then**
9:         $i_* \leftarrow i_* - 1$
10:     **end if**
11:     $d_{i_*} \leftarrow d_{i_*} + d_{\rho(1)}$   ▷ Update distance
12:     $\bar{\mathcal{D}} \leftarrow \bar{\mathcal{D}} \setminus \{d_{\rho(1)}\}$
13:     Recompute $\rho$, s.t., $d_{\rho(\ell)} \leq d_{\rho(\ell+1)}$
14: **end for**
15: $\hat{b} \leftarrow d_{\rho(1)}$
16: $\check{r} \leftarrow \check{b}/\hat{b}$

---

aims for finding prototype vectors. All three problems have known QUBO embeddings (Bauckhage et al., 2018; Biesner et al., 2022; Bauckhage et al., 2019). The specific setups are described in the Appendix.

### 6.1. Numerical Experiments

We compare different policies: the base policy $\dot{\pi}$ (baseline from (Mücke et al., 2025)), our B&B policy $\bar{\pi}_{\tilde{T}}$ with different rollout horizons $\tilde{T}$ and our rollout selection policy $\tilde{\pi}$. The relative DR reduction for a horizon up to $T = 10$ can be found in Fig. 6. We compare ALL (left) and IMPACT (right) for choosing the indices in the branch step. It is apparent that every single policy reduces the DR with an increasing horizon $T$. The base policy $\dot{\pi}$ is largely outperformed by our $\bar{\pi}_{\tilde{T}}$ and $\tilde{\pi}$. $\bar{\pi}_{\tilde{T}}$ is increasing its performance with an increasing rollout horizon $\tilde{T}$. We can see that the exact method ALL has the same performance as using the simplified version IMPACT, while being more computational demanding. It scales quadratically with the problem size $n$, where IMPACT is basically independent of $n$. The policy $\tilde{\pi}$ already almost achieves optimal performance (c.f. to $\bar{\pi}_{10}$).

Evaluating the quality of our bounds (see Sec. 5.3), we indicate the fraction of the pruned state space in Fig. 5. We here consider the exact solution, that is $\bar{\pi}_T$. We vary the depth until the upper bounds are updated. Increasing this depth, as well as increasing the horizon leads to pruning a larger fraction of the whole search space. The number of pruned states does not heavily depend on an updated current best, indicating the strength of our lower bound (Sec. 5.3).

### 6.2. Performance on Hardware Solvers

We first compare the DR reduction performance of the base policy $\dot{\pi}$ and a randomized base policy $\dot{\pi}^R$ with our rollout

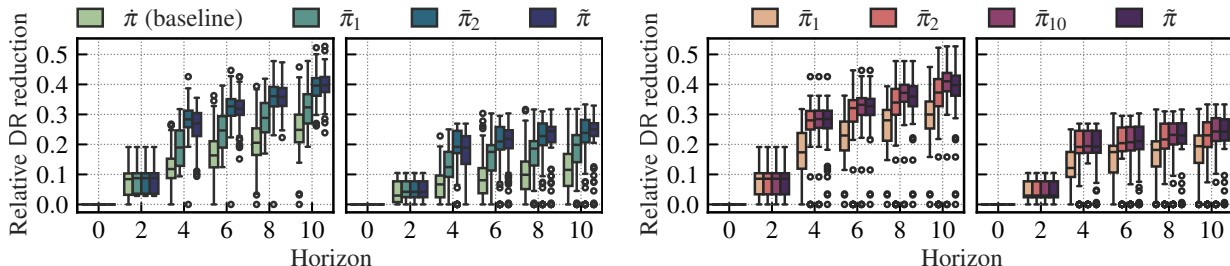

Figure 6: Relative DR reduction for 100 BINCLUS instances with $n = 8$ (first and third column plot) and $n = 16$ (second and fourth column plot). Different policies are compared for choosing the indices with ALL (left) and IMPACT (right).

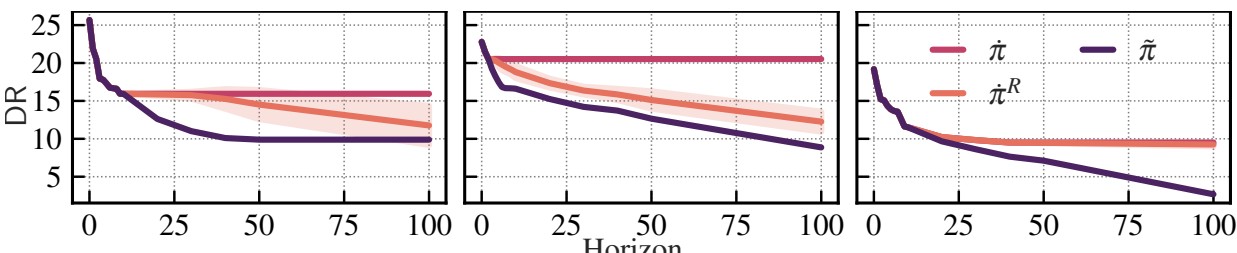

Figure 7: Performance of our developed policy $\tilde{\pi}$ compared to the base policy $\dot{\pi}$ and the randomized base policy $\dot{\pi}^R$. The DR reduction is compared for a SUBSUM (left), a BINCLUS (middle) and a VECQUANT (right) instance.

selection policy $\tilde{\pi}$ in Fig. 7 for three different instances of SUBSUM, BINCLUS and VECQUANT of dimensions $n = 16, 20, 20$. We can see that $\dot{\pi}$ ends up in local optima pretty fast while our $\tilde{\pi}$ is more robust and is steadily improving with an increasing horizon. It also outperforms $\dot{\pi}^R$, which needs a lot of iterations for an increasing QUBO dimension.

We also compare the results of our policies with two other methods from the literature. We denote the method of tuning penalty parameters for incorporating constraints in (Alessandroni et al., 2023) as PEN and the method of introducing auxiliary variables to reduce the BW (Oku et al., 2020) as AUX. However, they are not generally applicable to arbitrary QUBO problems, i.e., PEN can only be applied to VECQUANT since it incorporates a constraint of finding exactly $k$ prototypes and AUX can only be applied to SUBSUM, since we here use QUBO instances with integer values. However, our method can be applied to arbitrary QUBO problems. Specifics on the obtained DR can be found in Tab. 1—our method always achieves the smallest DR.

Further, we assess the impact of the reduced QUBO instances on two hardware QUBO solvers (Ising machines)—QA and DA. QAs are prone to *Integrated Control Errors* which constrain the DR of the hardware parameters. If the DR of the given QUBO is too large, it can happen that a completely different problem is solved due to implementation noise of the parameters. A similar problem appears for hardware solvers with a fixed bit precision: the parameters have to be rounded/quantized to that precision, which can lead to

different optima (see Fig. 1). Furthermore, DA designs can be improved by considering a reduced bit precision. In our experiments, the hardware plattform for the digital annealer is an AMD Virtex UltraScale+ FPGA VCU118 evaluation board. By implementing the digital annealer chip design with AMD Vivado for 16, 8 and 4 bit precision, we find that when using 4 instead of 16 bit precision, the number of on-chip signals is reduced by 28.29%. This has multiple benefits: First, the improvement allows us to run the orginal design with a reduced power consumption when operating the hardware solver. E.g., the power requirement for on-chip memory[1] shrinks from 1.4 W to 0.3 W. Second, due to less occupied chip space, the reduction also allows for an increase of the maximum number of QUBO problem variables. However we did not evaluate this option as it requires significant changes of the chip design which are out of scope of our study.

Now, after adressing the resource consumption, we answer the question whether reducing the DR leads to an optimized performance for QA and DA. Due to their probabilistic nature, we generate 1000 *samples* and use default parameters. For DA, we use three different bit precisions of the internal arithmetics, i.e., 16, 8 and 4 bit. For making the performances comparable we evaluate the

---

[1]Block RAM and Ultra RAM combined. See https://docs.amd.com/v/u/en-US/ug573-ultrascale-memory-resources. Numbers reported here are estimated by the FPGA design software.

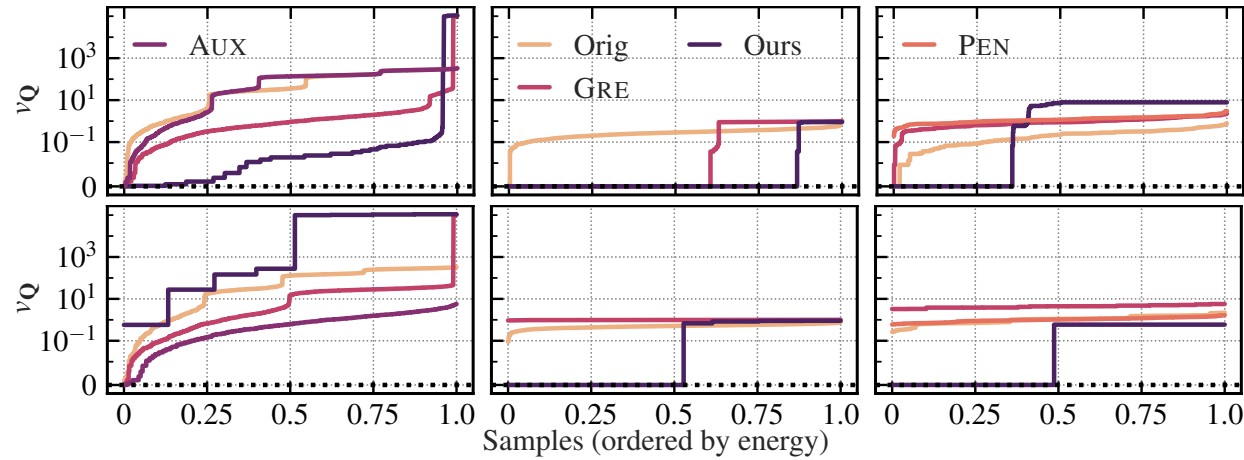

Figure 8: Performance of the D-Wave Advantage 5.4 (top row) and an FPGA-based digital annealer with 4-bit precision (bottom row): we compare the original QUBO $Q$ (Orig), the QUBO using the greedy base policy (GRE) for 100 steps $f_{100}(Q, \dot{\pi})$, the method in (Alessandroni et al., 2023) (PEN) for computing the penalty, the method of adding auxiliary variables (AUX) (Oku et al., 2020) and our rollout selection policy for 100 steps $f_{100}(Q, \tilde{\pi})$. The relative energies $v_Q$ for 1000 samples are shown for a SUBSUM (left), a BINCLUS (middle) and a VECQUANT (right) instance.

Table 1: Comparison of QUBO hardware solvers using different methods (details in Fig. 8). We depict the DR along with the number of optimal samples (from 1000) obtained by QA and DA with 16, 8 and 4 bit precision. Optima are bold and dashes indicate that the method is not applicable for the respective problem.

| | SUBSUM | | | | BINCLUS | | | | | VECQUANT | | | | |
|------|-------|-----|------|-----|-------|-----|------|------|------|--------|-----|------|------|-----|
| | DR | QA | DA16 | DA8 | DA4 | DR | QA | DA16 | DA8 | DA4 | DR | QA | DA16 | DA8 | DA4 |
| Orig | 25.68 | 0 | 5 | 1 | 0 | 22.79 | 3 | **1000** | **1000** | 1 | 19.19 | 18 | 1000 | 1000 | 0 |
| GRE | 15.94 | 3 | 687 | **3** | 0 | 20.52 | 605 | 462 | 0 | 0 | 9.51 | 1 | 1000 | 0 | 0 |
| AUX | 25.68 | 0 | 7 | 2 | **2** | — | — | — | — | — | — | — | — | — | — |
| PEN | — | — | — | — | — | — | — | — | — | — | 24.63 | 0 | 1000 | 1 | 0 |
| Ours | **9.89** | **26** | **1000** | 0 | 0 | **8.87** | **865** | 585 | 579 | **528** | **2.68** | **356** | **1000** | **1000** | **487** |

original QUBO energy $E_Q(z)$ for every sample $z$ and have a look at the relative distance to the optimum energy $v_Q(z) \coloneqq (E_Q(z) - E_Q(z^*))/E_Q(z^*)$. This is depicted in Fig. 8, where we indicate the energy distribution for the initial QUBO matrix $Q$, the base policy $f_{100}(Q, \dot{\pi})$, AUX for SUBSUM, PEN for VECQUANT and our rollout selection policy $f_{100}(Q, \tilde{\pi})$. Even though our DR reduction method can change the overall energy landscape, we are interested in the low energy values since we aim to minimize the energy. In Tab. 1, we depict the total number of optimal samples from the 1000 drawn samples, using different methods for QA and DA. We can see that our method almost always outperforms the baselines in terms of ability of the hardware solver to find optimal solutions.

## 7. Conclusion

In this paper, we developed a principled Branch-and-Bound algorithm for reducing the required precision for quadratic

unconstrained binary optimization problems (QUBO). In order to achieve this, we consider the dynamic range (DR) as a measure of complexity, which we iteratively reduce in a Markov decision process (MDP) framework. We propose computationally efficient and theoretically sound bounds for pruning, leading to drastic reduction of the search space size. Furthermore, we combine our approach with the well-known policy rollout for improving computational efficiency and the performance of already existing heuristics. Our extensive experiments comply with the theoretical insights. We use our method to reduce the DR of NP-hard real-word problems, such as clustering, subset sum and vector quantization. Our proposed algorithm largely outperforms recently developed algorithms, while also being applicable to arbitrary QUBO problems. The effectiveness for hardware solvers is shown for a real quantum annealer (QA) and an FPGA-based digital annealer (DA). We conclude that our method enhances the reliability of QA and DA in finding the optimum, and reduces the power consumption of DA.

## Impact Statement

This paper presents work whose goal is to advance the fields of Optimization and Artificial Intelligence. There are many potential societal consequences of our work, none which we feel must be specifically highlighted here.

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

## A. State Space Size

We proof the sub-exponential state space size given in Sec. 4.1 in the main paper. We have no knowledge about the final state $f_T(\boldsymbol{Q}, \pi)$ and thus the search space is exponentially large

$$\sum_{t=0}^{T} (n^2)^t = \frac{(n^2)^{T+1} - 1}{n^2 - 1} \in \mathcal{O}(n^{2(T+1)}) . \tag{8}$$

Choosing the number of iterations $T$ logarithmic in the problem dimension, i.e., $T = \log_2(n^2) = 2\log_2(n)$, results in the sub-exponential state space size of

$$\mathcal{O}\left(n^{4\log_2(n)+2}\right) = \mathcal{O}\left(n^2 2^{\left(4\log_2^2(n)\right)}\right) \subseteq o(2^n) .$$

Thus, we have an asymptotically slower growth than the exponentially large state space size $2^n$ of the original QUBO problem.

## B. Details on Boundaries and Heuristic

We recall how the boundaries $y^-(\boldsymbol{Q}) \leq w \leq y^+(\boldsymbol{Q})$ from Sec. 4.1 are computed in (Mücke et al., 2025). Let $\mathbb{B} := \{0, 1\}$, $\boldsymbol{z} \in \mathbb{B}^n$ and $k, l \in [n]$. The $n$-dimensional vector which variables indexed by $k$ and $l$ fixed to the values $(a, b) \in \mathbb{B}^2$ is indicated by $\boldsymbol{z}_{ab}$. Then

$$\boldsymbol{z}_{ab}^* \in \operatorname*{arg\,min}_{\boldsymbol{z}_{ab} \in \{0,1\}^n} E_{\boldsymbol{Q}}(\boldsymbol{z}_{ab}) ,$$

and define $y_{ab}^* := E_{\boldsymbol{Q}}(\boldsymbol{x}_{ab}^*)$. Naturally, $y_{ab}^*$ are just as hard to compute as solving QUBO itself. Therefore, we work with upper and lower bounds on the true values to determine the update parameter $w$, which are much easier to compute. Bounds for $y_{ab}^*$ are denoted by $\hat{y}_{ab}$ and $\check{y}_{ab}$, such that

$$\check{y}_{ab} \leq y_{ab}^* \leq \hat{y}_{ab}, \ \forall(a, b) \in \mathbb{B}^2 .$$

Further, let

$$y^- := \min\{0, \min\{\hat{y}_{00}, \hat{y}_{01}, \hat{y}_{10}\} - \check{y}_{11}\} ,$$
$$y^+ := \max\{0, \min\{\check{y}_{00}, \check{y}_{01}, \check{y}_{10}\} - \hat{y}_{11}\} ,$$

if $k \neq l$. Otherwise, when $k = l$, let $y^- = \min\{0, \hat{y}_{00} - \check{y}_{11}\}$ and $y^+ = \max\{0, \check{y}_{00} - \hat{y}_{11}\}$. Then $\boldsymbol{Q} + w\boldsymbol{e}_k\boldsymbol{e}_l^\top \trianglelefteq \boldsymbol{Q}$ if (Theorem 1 in (Mücke et al., 2025))

$$y^- \leq w \leq y^+ .$$

For the proof, we refer to the paper. The lower bound $\check{y}$ can be efficiently computed, e.g. with roof-duality (Boros et al., 2008) and an upper bound $\hat{y}$ with a local optimum obtained through any QUBO solver.

With the boundaries in Appendix B at hand, we have an interval in which a QUBO weight can be changed to preserve

an optimizer. Instead of considering the weight update as another degree of freedom, we use the best performing heuristic from (Mücke et al., 2025). It is a greedy strategy where the QUBO parameter $Q_{kl}$ is increased if $q_\ell < 0$ and decreased otherwise, where $\sigma(\ell) = (k, l)$. For increasing (decreasing) $Q_{kl}$, $w$ is chosen maximally (minimally) while not increasing the DR. Recall that there is always a $q_u = 0$ for some $u \in [m]$, and thus we may set parameters to 0. For certain target platforms, such as quantum annealers, this is particularly beneficial, as setting a parameter to 0 allows to discard the coupling between the qubits indexed by $k$ and $l$, which saves hardware resources. Thus, we always set parameters to 0 if possible. More technical details on the used heuristic can be found in (Mücke et al., 2025).

## C. Complexity Analysis

The following complexity analysis belongs to computing the bounds in Sec. 5.3 of the main paper. Even though we are able to prune a large amount of the search space, it would be beneficial for the bounds to be computable efficiently. It turns out that the bounds can be dynamically computed in $\mathcal{O}(Tn^2)$. Through the use of memoization, this can be efficiently combined with PR.

**Upper Bound** For the transition $f(\boldsymbol{Q}, (k, l)) = \boldsymbol{Q} + h(\boldsymbol{Q}, (k, l))\boldsymbol{e}_k\boldsymbol{e}_l^\top$ we need to compute the value $h(\boldsymbol{Q}, (k, l))$ which is implicitly dependent on $y^-(\boldsymbol{Q})$ and $y^+(\boldsymbol{Q})$. For computing a lower bound $\check{y}$, we use the roof dual technique (Boros et al., 2008). A flow network is build with $\mathcal{O}(n)$ nodes, and the lower bound is given by the maximum flow value, which is computable in $\mathcal{O}(n^3)$. Exploiting the top-down nature of our B&B approach, we can dynamically update the flow network and recompute the maximum flow in $\mathcal{O}(n^2)$. An upper bound $\hat{y}$ is given by performing local descent using a discrete analogue of a gradient (Boros et al., 2006). Having an initial runtime of $\mathcal{O}(n^3)$ it also can be dynamically updated making it computable in $\mathcal{O}(n^2)$. This leads to an initial computational effort of $O(n^3)$ and $O(n^2)$ for every subsequent branched state.

Having $y^-(\boldsymbol{Q})$ and $y^+(\boldsymbol{Q})$ at hand, $h$ can be computed in $\mathcal{O}(n)$. This leads to a total computational cost of $\mathcal{O}(n^2)$ for a single state. Thus, every upper bound $\hat{r}(\boldsymbol{Q}, T)$ (policy rollout) can be computed in $\mathcal{O}(Tn^2)$.

**Lower Bound** For the computational complexity of the lower bound $\check{r}(\boldsymbol{Q}, T)$, we first consider the lower bound $\check{b}(\boldsymbol{Q}, T)$. Initially, the parameters of $\boldsymbol{Q}$ and the elements in $\bar{\mathcal{D}}$ can be sorted in $\mathcal{O}(n^2 \log(n))$. Updating single parameters, this sorting can be dynamically updated in $\mathcal{O}(n^2)$. For $\hat{b}(\boldsymbol{Q}, T)$, this is repeated $T$ times, leading to a computational effort of $\mathcal{O}(Tn^2)$. The bound $\check{b}(\boldsymbol{Q}, T)$ can be computed in $\mathcal{O}(T)$. Thus, the computational complexity of

the lower bound $\check{r}(\boldsymbol{Q}, T)$ is $\mathcal{O}(Tn^2)$. Combined with the runtime for computing an upper bound $\hat{r}(\boldsymbol{Q}, T)$ results in a total runtime $\mathcal{O}(Tn^2)$ of the bound-step.

# D. Data Generation

The following setup for the datasets are used in Sec. 6 in the main paper.

### D.1. BINCLUS

To generate data for BINCLUS, we begin by sampling $n$ independent and identically distributed 2-dimensional points from an isotropic normal distribution $\mathcal{N}(0, 0.1)$. Next, we create two clusters by transforming the first $n/2$ points with $(x_1, x_2) \mapsto (x_1 - 1, x_2)$ and the last $n/2$ points with $(x_1, x_2) \mapsto (x_1 + 1, x_2)$. Finally, we select the first and last points and multiply their coordinates by 100, introducing two outliers into the data set. For analyzing the behaviour of our B&B algorithm, we sample 100 BINCLUS QUBO instances for $n = 8, 16$.

A corresponding QUBO embedding is given as follows: Assume we are given a set of $n$ data points $\mathcal{X} \subset \mathbb{R}^d$, $|\mathcal{X}| = n$. We want to partition $\mathcal{X}$ into disjoint clusters $\mathcal{X}_1, \mathcal{X}_2 \subset \mathcal{X}$, $\mathcal{X}_1 \dot\cup \mathcal{X}_2 = \mathcal{X}$. We gather the data in a matrix $\boldsymbol{X} := \begin{bmatrix} \boldsymbol{x}^1, \ldots, \boldsymbol{x}^n \end{bmatrix}, \forall i : \boldsymbol{x}^i \in \mathcal{X}$, and assume that it is centered, i.e., $\boldsymbol{X}\mathbf{1} = \mathbf{0}$, where $\mathbf{1}$ denotes the $n$-dimensional vector consisting only of ones. A QUBO formulation was derived in (Bauckhage et al., 2018), which minimizes the *within cluster scatter*:

$$\min_{\boldsymbol{s} \in \{-1, +1\}^n} - \boldsymbol{s}^\top \boldsymbol{X}^\top \boldsymbol{X} \boldsymbol{s} \tag{10a}$$

$$\Leftrightarrow \min_{\boldsymbol{z} \in \{0,1\}^n} - \boldsymbol{z}^\top \boldsymbol{X}^\top \boldsymbol{X} \boldsymbol{z} + \mathbf{1}^\top \boldsymbol{X}^\top \boldsymbol{X} \boldsymbol{z} \,, \tag{10b}$$

where $\boldsymbol{s} = 2\boldsymbol{z} - \mathbf{1}$. A value $z_i = 1$ indicates that data point $\boldsymbol{x}^i$ is in cluster $\mathcal{X}_1$, and in $\mathcal{X}_2$ for $z_i = 0$. Observing that $\boldsymbol{X}^\top \boldsymbol{X}$ is a Gram matrix leads to a possible application of the kernel trick. For this, we consider a centered kernel matrix $\boldsymbol{K} \in \mathbb{R}^{n \times n}$ with elements $k(\boldsymbol{x}^i, \boldsymbol{x}^j)$, where $k : \mathbb{R}^n \times \mathbb{R}^n \to \mathbb{R}$ is a kernel function. $k(\boldsymbol{x}^i, \boldsymbol{x}^j)$ indicates how similar data points $\boldsymbol{x}^i$ and $\boldsymbol{x}^j$ are in some feature space. We can reformulate (10b) to

$$\min_{\boldsymbol{z} \in \{0,1\}^n} \mathbf{1}^\top \boldsymbol{K} \boldsymbol{z} - \boldsymbol{z}^\top \boldsymbol{K} \boldsymbol{z} \,,$$

which gives us our QUBO formulation. For our experiments, we choose a linear kernel. For evaluating the hardware solver performance, we use the same instance as in (Mücke et al., 2025).

### D.2. SUBSUM

For the SUBSUM problem, we are given a set $\mathcal{A} = \{a_1, \ldots, a_n\} \subset \mathbb{Z}$ and $T \in \mathbb{Z}$. The goal is to find $\mathcal{I} \subset [n]$,

s.t., $\sum_{i \in \mathcal{I}} a_i = T$. We use the same problem instance as is given in (Mücke et al., 2025), Section 4.2. We set $n = 16$ and generate the elements of $\mathcal{A}$ as $\lfloor |10 \cdot Z| \rfloor$, where $Z$ follows a standard Cauchy distribution. This approach leads to occasional outliers with large magnitudes, which in turn produced QUBO instances with a high degree of difficulty due to large dynamic ranges (DR). Next, we determined the number of summands $k$ by sampling from a triangular distribution $\lfloor U \rfloor$, where $U$ is defined with parameters $a = \frac{n}{5}$, $b = \frac{n}{2}$, and $c = \frac{4n}{5}$, ensuring that, on average, half of the elements of $\mathcal{A}$ contribute to the sum. Finally, we selected $k$ indices from $[n]$ without replacement to form the subset $\mathcal{I}$ and set $T = \sum_{i \in I} a_i$, thereby creating problems where the global optimum is predetermined.

We obtain a QUBO formulation, where we use $n$ binary variables which indicate if $i \in S$ for each $i$. With $\boldsymbol{a} = (a_1, \ldots, a_n)$, a QUBO formulation is given by

$$\min_{z \in \{0,1\}^n} \left( \boldsymbol{a}^\top \boldsymbol{z} - T \right)^2 \Leftrightarrow \min_{z \in \{0,1\}^n} \boldsymbol{z}^\top \boldsymbol{a}^\top \boldsymbol{a} \boldsymbol{z} - 2T \boldsymbol{a}^\top \boldsymbol{z} \,.$$

### D.3. VECQUANT

Vector quantization deals with the problem of finding prototypes of a given set of vectors, which give a best representation according to some measure. We use the approach from (Bauckhage et al., 2019), where the goal is to find $k$ medoids according to the well known $k$-medoids objective function. A QUBO formulation is given by

$$\min_{\boldsymbol{z} \in \{0,1\}^n} \boldsymbol{z}^\top \left( \gamma \mathbf{1}\mathbf{1}^\top - \alpha \boldsymbol{D} \right) \boldsymbol{z} + \left( \beta \boldsymbol{D}\mathbf{1} - 2\gamma k\mathbf{1} \right)^\top \boldsymbol{z} \,,$$

where $\boldsymbol{D}$ is a pairwise distance matrix for, $\alpha$ is a weight for identifying far apart data points, $\beta$ is for identifying central data points and $\gamma$ ensures that we choose exactly $k$ vectors. We follow (Bauckhage et al., 2019) and set $\alpha = 1/k$, $\beta = 1/n$ and use Welsh's distance

$$d(\boldsymbol{x}, \boldsymbol{y}) := 1 - \exp\left( -\frac{\|\boldsymbol{x} - \boldsymbol{y}\|_2^2}{2} \right) \,,$$

for computing $\boldsymbol{D}$. The penalty parameter $\gamma$, which enforces that exactly $k$ prototypes are chosen, is set to 2. For hardware evaluation, we use the same dataset as BINCLUS and set $k = 4$.

# E. Baselines

We compare our approaches to different baseline methods from the literature.

### E.1. Adding Auxiliary Variables

In (Oku et al., 2020), a method is proposed which reduces the bit-width of single parameters of the underlying Ising

model. It adds auxiliary variables to the problem in a way such that an optimal configuration still corresponds to an optimum of the original problem. It is assumed that all problem parameters are integer, i.e., $\boldsymbol{J} \in \mathbb{Z}^{n \times n}, \boldsymbol{h} \in \mathbb{Z}^n$. If we want to change a parameter $J_{ij}, i \neq j$ to reduce its bit-width, we can do this by introducing a new variable $x$ to the problem to obtain $\boldsymbol{J}' \in \mathbb{Z}^{n+1 \times n+1}$ such that

$$J'_{ij} = J_{ij} - r, \ J'_{ix} = |r|, \ J'_{xj} = -r \ .$$

If we want to change $h_i$, optimality is ensured by

$$h'_i = h_i - r, \ h'_{ix} = -|r|, \ h'_x = r \ .$$

That is, if $\bar{\boldsymbol{s}} := \arg\min_{\boldsymbol{s} \in \{-1,1\}^{n+1}} \boldsymbol{s}^\top J' \boldsymbol{s} + \boldsymbol{s}^\top \boldsymbol{h}'$, then

$$\bar{\boldsymbol{s}}_{[n]}^\top \boldsymbol{J} \bar{\boldsymbol{s}}_{[n]} + \bar{\boldsymbol{s}}_{[n]}^\top \boldsymbol{h} = \min_{\boldsymbol{s} \in \{-1,+1\}^n} \boldsymbol{s}^\top J \boldsymbol{s} + \boldsymbol{s}^\top \boldsymbol{h} \ ,$$

where the subscript $[n]$ denotes the vector consisting of the first $n$ entries. However, to reduce the bit-width by $m$ bits, $\mathcal{O}(n2^m)$ auxiliary variables are introduced. Due to limited capability of hardware solvers in terms of representable problem size, this can pose a problem on finding a solution.

In our experiments we apply this method to SUBSUM, since our datasets are integer. We reduce the bit-width of our parameters by 2, since no further benefit on the performance of hardware solvers was observed for introducing more variables.

### E.2. Tuning Penalty Parameters

As a second baseline, we use the method from (Alessandroni et al., 2023). It is assumed that the optimization problem is given in a quadratic binary constrained form

$$\min_{\boldsymbol{z} \in \{0,1\}^n} \boldsymbol{z}^\top \boldsymbol{Q} \boldsymbol{z}$$
$$\text{s.t. } \boldsymbol{A}\boldsymbol{z} = \boldsymbol{b} \ ,$$

which can be brought in an equivalent QUBO form by introducing a penalty parameter $\lambda > 0$

$$\min_{\boldsymbol{z} \in \{0,1\}^n} \boldsymbol{z}^\top \boldsymbol{Q} \boldsymbol{z} + \lambda(\boldsymbol{A}\boldsymbol{z} - \boldsymbol{b})^\top(\boldsymbol{A}\boldsymbol{z} - \boldsymbol{b}) \ .$$

This parameter has to be chosen large enough to ensure equivalence. We use the method

$$\lambda = \hat{\boldsymbol{z}}^\top \boldsymbol{Q} \hat{\boldsymbol{z}} - \check{E}_{\boldsymbol{Q}} \ ,$$

where $\hat{\boldsymbol{z}}$ is a feasible solution, i.e., $\boldsymbol{A}\hat{\boldsymbol{z}} = \boldsymbol{b}$ and $\check{E}_{\boldsymbol{Q}}$ is a lower bound on the optimum, i.e., $\check{E}_{\boldsymbol{Q}} \leq \min \boldsymbol{z}^\top \boldsymbol{Q} \boldsymbol{z}$. For increasing $\lambda$ the DR is also increased, thus choosing $\lambda$ as small as possible is favourable. Hence, we use the exact solutions for computing $\lambda$ in our experiments. However, this is intractable in realistic scenarios.

