# OpenReview forum: "Dynamic Range Reduction via Branch-and-Bound"
_ICML.cc/2025/Conference — Submitted to ICML 2025_

### Official Review · Reviewer_MyTu · 2025-03-13

**Overall Recommendation:** 3

**Summary:**

This paper tackles the numerical precision challenges of solving NP-hard QUBO problems on low-precision hardware accelerators (e.g., quantum annealers, FPGAs) by introducing a dynamic range (DR)-aware optimization framework. The authors propose a hybrid Branch-and-Bound algorithm with policy rollout to iteratively compress the DR of QUBO matrices while preserving global optima, enabling compatibility with reduced-precision representations. Key innovations include formalizing DR reduction as a Markov decision process, designing efficient bounds for pruning suboptimal search paths, and validating the method on real hardware. Experiments demonstrate significant DR reduction across ML-related QUBO instances, outperforming greedy baselines and enhancing solvability on quantum/FPGA platforms.

**Claims And Evidence:**

Yes, the claims are supported by clear and convincing evidence.

**Essential References Not Discussed:**

No critical references are omitted.

**Experimental Designs Or Analyses:**

Experimental designs are sound but could be expanded. BinClus/SubSum datasets rely on synthetic outliers (Appendix D), raising concerns about real-world applicability.

**Methods And Evaluation Criteria:**

The methods and evaluation criteria are appropriate but require scalability validation. For example, experiments are limited to small-scale QUBO instances with n≤20 (i.e., 20 binary variables). Scalability for large-scale problems with n>100 remains unverified.

**Other Comments Or Suggestions:**

N/A

**Other Strengths And Weaknesses:**

Strengths:

(1) Originality: First to formalize DR reduction as an MDP and integrate B&B with policy rollout for QUBO.

(2) Practical impact: Validates DR compression on real hardware, enabling low-precision AI accelerators.

Weaknesses:

(1) Theoretical gaps: No convergence guarantees for policy rollout.

(2) Limited scalability: Experiments focus on small-scale problems (n≤20).

**Questions For Authors:**

Q1: How does the method guarantee global optima preservation when the initial z∗ is unknown (e.g., for high-dimensional QUBO)?

Q2:  How sensitive is the method to the choice of the rollout depth and the number of iterations?

Q3: While the paper focuses on hardware solvers (QA and DA), classical solvers are often used for QUBO problems. How does the proposed DR reduction method compare to classical solvers in terms of solution quality and runtime for the same QUBO instances?

**Relation To Broader Scientific Literature:**

The work extends prior research in meaningful ways:

1. DR as a hardware-centric metric: Builds on Stollenwerk et al. (2019a,b) and Yachi et al. (2023) but generalizes beyond CR/BW.
2. Algorithmic innovation: Integrates policy rollout (Bertsekas et al., 1997) into B&B, addressing local optima in Mücke et al. (2025).

**Theoretical Claims:**

Yes, the proofs for theoretical claims is correct.

---

> ### Author Rebuttal · Authors · 2025-03-31
>
> We thank the reviewer for the thoughtful and detailed feedback, as well as for highlighting the originality, theoretical soundness, and practical relevance of our contributions. We respond to the raised concerns below.
>
> ### Scalability and Small-Scale Evaluation ($n \le 20$)
>
> We acknowledge that the experiments focus on QUBO instances with up to 20 binary variables. This is primarily due to hardware limitations of current quantum solvers, which restrict instance size due to limited qubit count or precision granularity. Nonetheless, we emphasize that:
>
> - Our algorithm is not limited to small-scale instances in principle.
> - The computational complexity is governed by the most time consuming step of computing bounds in the Branch-and-Bound (B&B) algorithm. The runtime is $O(Tn^2)$, where $T$ is the number of parameters we allow to change, and $n$ the problem dimensionality. In fact, as shown in the experiments, even modifying a few parameters ($T\ll n$) can result in substantial dynamic range (DR) reduction.
> - We are currently extending our evaluation with larger-scale synthetic and structured QUBO instances and will include this in the final version to validate scalability in software simulations.
>
> ### Synthetic Nature of Datasets and Real-World Applicability
>
> The BINCLUS and SUBSUM datasets indeed contain synthetic outliers to simulate worst-case DR conditions, which are commonly encountered in QUBO formulations derived from noisy or high-variance data (e.g., weighted constraints, learned potentials).
> We agree that real-world benchmarks (e.g., Max-Cut QUBOs) can further validate applicability. We are currently incorporating additional benchmarks and realistic QUBO instances into our extended evaluation.
>
> ### Convergence Guarantees of Policy Rollout
>
> We appreciate the reviewer’s observation. Policy rollout introduces a trade-off between solution quality and computational complexity:
> - Since rollout is performed for a fixed finite depth, the search is guaranteed to terminate.
> - Moreover, rollout guarantees an improvement over the base (greedy) policy in terms of DR. We will clarify this guarantee in the final version and discuss potential directions for formal convergence analysis.
>
> ### Q1: Global Optima Preservation without Knowing z*?
>
> We believe this might stem from a misinterpretation. Our algorithm does not assume prior knowledge of the optimal solution z*. Rather, we guarantee preservation of some global optimum by building on (Mücke et al., 2025), which defines safe parameter intervals—i.e., updates that provably preserve at least one global optimum of the original QUBO.
> These intervals are derived using efficiently computable bounds on the optimal QUBO value, as discussed in Appendix B:
> - Upper bounds are obtained via approximation algorithms (e.g., simulated annealing).
> - Lower bounds are computed efficiently via roof duality (Boros et al., 2008) or semidefinite relaxations (Alessandroni et al., 2023).
>     This framework allows us to conservatively modify QUBO parameters without altering the problem's global optima.
>
> ### Q2: Sensitivity to Rollout Depth and Iteration Horizon
>
> Our experiments (Fig. 6) demonstrate that larger rollout depths and iteration counts yield stronger DR reduction. However, our algorithm remains efficient due to:
> - Early termination via policy rollout, limiting search depth.
> - Impact-based parameter selection (IMPACT), which restricts updates to DR-relevant matrix entries, reducing the branching factor.
> This enables us to scale to larger horizons T while maintaining computational tractability. We will add a sensitivity analysis to the final version to further illustrate this.
>
> ### Q3: Comparison to Classical Solvers
>
> We analyzed simulated annealing (Kirkpatrick et al., 1983) and observed that lower DR does not necessarily correlate with improved classical solver performance—likely due to classical methods being less sensitive to DR than hardware-based solvers due to floating-point arithmetics.
> We will incorporate a detailed comparison with simulated annealing and tabu search (Goldberg & Kuo, 1987) in the final version to provide a more complete picture of solver performance post-DR reduction.
>
> We thank the reviewer again for the insightful suggestions, which will directly inform the improvements in our final submission.

---

> > ### Comment · Reviewer_MyTu · 2025-04-04
> >
> > Thank the authors for the response. Many of my concerns are addressed, so I would like to raise my score to 3.

---

### Official Review · Reviewer_UsMc · 2025-03-14

**Overall Recommendation:** 2

**Summary:**

For given QUBO instances, the presented approach produces new QUBO instances which feature the same solutions but whose parameters have a reduced dynamic range. This is achieved by formulating the problem as an MDP and running a branch-and-bound strategy. Results show an improved number of found global optima over several runs on three different types of problems.

**Claims And Evidence:**

The claims made are sound. The generality of the observed phenomenon is to be questioned and not further discussed, i.e., the title could use an addition like "in X cases".

**Essential References Not Discussed:**

see above

**Experimental Designs Or Analyses:**

The experimental design lacks a broader evaluation for different kinds of problem instances. It is also lacking any comparison to non-MDP-based approaches to the same issue or even any comparison to simpler heuristics tackling the same problem. (D-Wave, for example, per default applies some manipulation on the weights to adjust dynamic range.) The setup of the quantum hardware was described insufficiently in this case.

**Methods And Evaluation Criteria:**

The chosen problem set is very narrow. QUBO problems with less sensitivity to dynamic range (e.g., QUBO problems with a fixed dynamic range) are not discussed. General (native) QUBO instances are not discussed. The approach is not intently built for these kinds of problems, but it would still be helpful to show the results for them.

**Other Comments Or Suggestions:**

typos:
- refer to equation with "Equation 1" not "(1)"
- refer to literature via citet and citep depending on context
- use of "NP-hard" and "complexity" is a bit imprecise

**Other Strengths And Weaknesses:**

see above

**Questions For Authors:**

none

**Relation To Broader Scientific Literature:**

The paper is missing some discussion about other heuristics for the dynamic range problem.

**Theoretical Claims:**

The main theoretical claim is the construction of the upper and lower bounds. They appear fine at first glance.

---

> ### Author Rebuttal · Authors · 2025-03-31
>
> We thank the reviewer for the detailed and constructive review. Below, we address the key concerns and clarify aspects related to the scope, comparisons, and experimental setup. We will incorporate the suggested corrections (e.g., references to equations, citation style, and terminology) in the camera-ready version.
>
> ### Scope of QUBO Problem Classes Evaluated
>
> Our current focus is on QUBO problems with high dynamic range due to input data, which are particularly challenging for real-world hardware solvers. We agree that extending the evaluation to QUBO problems with lower sensitivity to DR and more general or randomly generated instances is important. We are actively working on expanding our experiments to cover a broader spectrum of QUBO classes and will integrate this extended evaluation into the final version.
>
> ### Comparison to Non-MDP Heuristics and Literature Coverage
>
> Our experimental section includes comparisons to several established approaches to dynamic range reduction that preserve global optima:
> - The greedy heuristic from (Mücke et al., 2025)
> - The AUX method using auxiliary variables (Oku et al., 2020)
> - The PEN method for tuning penalty parameters (Alessandroni et al., 2023)
>
> These baselines represent different families of strategies. If the reviewer has a specific alternative heuristic in mind, we would be happy to include a comparison in the final version and expand the discussion on related dynamic range mitigation strategies, particularly those used in practice by hardware vendors.
>
> ### Clarification on D-Wave’s Built-in Techniques
>
> We appreciate the note regarding D-Wave’s internal dynamic range adjustments. While D-Wave applies internal rescaling and chain strength tuning to mitigate embedding-related issues, we note:
> - Global parameter rescaling does not change the dynamic range.
> - Their techniques are hardware-specific (focused on embedding quality and chain robustness), whereas our method operates on the QUBO formulation itself and is hardware-agnostic. We will clarify this distinction in the final version and note that a deeper integration with D-Wave’s toolchain is a promising direction for future work.
>
> ### Quantum Hardware Setup Details
>
> In our quantum experiments, we used D-Wave’s Advantage 5.4 system with default solver parameters. We evaluated results over 1000 samples per QUBO instance and reported energy distributions. We agree that additional parameters (e.g., annealing time, number of reads) can improve transparency and will include these in the updated paper.
>
> Once again, we appreciate the reviewer’s insights. We will integrate the broader evaluations, more detailed hardware setup, and improved contextualization in the camera-ready version.

---

### Official Review · Reviewer_PJ4k · 2025-03-16

**Overall Recommendation:** 3

**Summary:**

This paper presents a Branch-and-Bound algorithm designed to reduce the numerical precision requirements of NP-hard Quadratic Unconstrained Binary Optimization (QUBO) problems, which are critical in real-time AI applications. By utilizing dynamic range as a measure of complexity, the algorithm aims to enhance the solvability of QUBO problems on hardware accelerators like quantum and FPGA-based digital annealers. The experimental results demonstrate that the proposed method effectively reduces the dynamic range in problems such as subset sum, clustering, and vector quantization.

**Claims And Evidence:**

Yes, the claims are supported by both theoretical analysis and empirical experiments.

**Essential References Not Discussed:**

The introduction to related work is relatively thorough.

**Experimental Designs Or Analyses:**

Yes, the experiment section is structured.

**Methods And Evaluation Criteria:**

Yes, this work brings a new perspective and method to Quadratic Unconstrained Binary Optimization.

**Other Comments Or Suggestions:**

None

**Other Strengths And Weaknesses:**

1. The author provides insufficient information regarding data input and does not clarify what QUBO embedding is. Additionally, the appendix does not include any statistical details about the dataset.

2. The experiment appears to be somewhat inadequate, as the information provided in Table 1 is limited. AUX and PEN do not work in most cases, and the author should conduct a more comprehensive evaluation.

**Questions For Authors:**

1. The author provides insufficient information regarding data input and does not clarify what QUBO embedding is. Additionally, the appendix does not include any statistical details about the dataset.

2. The experiment appears to be somewhat inadequate, as the information provided in Table 1 is limited. AUX and PEN do not work in most cases, and the author should conduct a more comprehensive evaluation.

**Relation To Broader Scientific Literature:**

This paper aims to solve the Quadratic Unconstrained Binary Optimization issue that can not be solved by the existing works, which is a further improvement on the existing methods.

**Theoretical Claims:**

Yes, the theoretical proofs in the manuscript are solid.

---

> ### Author Rebuttal · Authors · 2025-03-31
>
> We thank the reviewer for the positive and encouraging evaluation of our paper, and for acknowledging the strength of our theoretical and empirical contributions. Below, we address the noted concerns regarding dataset details and experimental evaluation.
>
> ### Clarification on Data Input and QUBO Embedding
>
> We appreciate the reviewer’s feedback on the clarity of the data input process. While the Appendix includes a description of the datasets, we agree that this could be made more explicit. In the camera-ready version, we will:
> - Expand the descriptions of the BINCLUS, SUBSUM, and VECQUANT problems.
> - Clearly define the QUBO embeddings used—i.e., how each problem is reformulated into a QUBO structure. For example, the 2-means clustering task is translated into a QUBO that optimizes a discrete assignment of points to clusters.
>
> ### Lack of Statistical Details in the Appendix
>
> This is a valid and helpful point. While our current focus was on reduction of dynamic range and hardware performance, we agree that statistical properties of the input data can influence QUBO structure and dynamic range. We will include:
> - Summary statistics (e.g., dimensionality, distributional properties, sparsity) for each dataset.
> - A brief discussion of how these properties relate to dynamic range and solver performance.
>
> ### Concerns About Table 1 and Baseline Methods (AUX, PEN)
>
> We acknowledge that AUX and PEN do not apply to all QUBO instances. This reflects their inherent limitations in generality:
> - AUX is only suitable for integer-valued QUBOs, and is therefore restricted to problems like SUBSUM.
> - PEN is applicable only when penalty parameters are used to enforce hard constraints (e.g., in VECQUANT).
>
> In contrast, our method is universally applicable to any real-valued QUBO, making it suitable across problem domains without requiring specific problem structure. We will clarify this in the final version.
>
> ### Request for More Comprehensive Evaluation
>
> We fully agree that a broader empirical evaluation is valuable. While we selected three representative problems (subset sum, clustering, vector quantization), we are already extending our evaluation to include additional QUBO formulations and larger problem sizes in preparation for the final version.
>
> We sincerely thank the reviewer for the helpful suggestions. We will incorporate these improvements in the camera-ready version to ensure clarity and completeness.

---

### Official Review · Reviewer_ZLAG · 2025-03-22

**Overall Recommendation:** 2

**Summary:**

The focus of this paper is the Quadratic Unconstrained Binary Optimization problem (QUBO), and in particular on methods to reduce the precision of the input entries. This is motivated by applications in hardware acceleration, where small input (e.g. 8 bits) can result in better parallelization. QUBO is an NP-hard problem, and can model many combinatorial optimization problems. They proposed several principled methods of dealing with this issue, including a branch and bound algorithm. Finally, an experimental evaluation is included on quantum hardware and FPGA-based digital annealer.

**Claims And Evidence:**

I did not see any problematic claims.

**Essential References Not Discussed:**

I did not notice essential references not discussed, but am not familiar with related work on this topic.

**Experimental Designs Or Analyses:**

The experimental design and analyses seemed okay to me, but again I am not an expert on these applications.

**Methods And Evaluation Criteria:**

They seemed to make sense to me, but I am not an expert on the standards for this particular area.

**Other Comments Or Suggestions:**

None

**Other Strengths And Weaknesses:**

Strengths
- The paper was well-written and easy for me to read. It seemed polished.
- The QUBO problem seemed very general, and it is stated in the paper that it could model a large variety of problems in combinatorial optimization.
- I think it's interesting to consider the more low level runtime considerations in combinatorial optimization, like how the input is represented in bits.

Weaknesses
- The presented contributions were principled, but they were not backed up with theoretical proof. The paper describes a bad alternative to the issue of reducing precision as simply truncating the input entries since that could completely change the CO problem, but since there doesn't seem to be proof that their way will preserve the problem I don't see how we can be confident that their way won't also completely change the CO problem.
- In addition to the fact that the algorithmic results are principled but not backed up by rigorous analysis, it didn't seem like there was a ton of novel results compared to other comparable papers I've seen at ICML.
- It seems that the presented algorithm is exponential time, but if we are later going to run an approximation algorithm for the CO problem wouldn't the runtime for the precision procedure be a prohibitive bottleneck?

**Questions For Authors:**

- Could you clarify what can be said for sure about the performance of the algorithm? In terms of theoretical guarantees.

**Relation To Broader Scientific Literature:**

This paper may be relevant to the hardware acceleration community. I don't think that the majority of researchers in combinatorial optimization would be interested in the results, but there may be some.

**Theoretical Claims:**

The algorithms and analysis were described as principled, as opposed to ones with rigorous theoretical guarantees.

---

> ### Author Rebuttal · Authors · 2025-03-31
>
> We thank the reviewer for the thoughtful and constructive feedback. Below, we respond to the raised concerns regarding theoretical guarantees, runtime feasibility, and novelty.
>
> ### Theoretical Guarantees and Rigor
>
> While the overall optimization procedure is heuristic in nature, key components of our method are theoretically grounded:
>
> #### Bounding Procedure:
> Our Branch-and-Bound (B&B) algorithm leverages rigorously derived lower and upper bounds on the dynamic range (DR) that can be achieved through permissible modifications to the QUBO matrix. These bounds are mathematically valid and discussed in detail in Section 5.3 and Appendix C.
>
> #### Preservation of Optima:
>
> As noted, naive truncation of QUBO parameters can lead to incorrect solutions (e.g., spurious optima). In contrast, we build upon (Mücke et al., 2025), which defines intervals for safe parameter updates—i.e., updates that provably preserve at least one global optimum of the original QUBO.
> These intervals are derived using efficiently computable bounds on the optimal QUBO value, as discussed in Appendix B:
>
> - Upper bounds via sub-optimal solutions (e.g., simulated annealing).
> - Lower bounds via roof duality (Boros et al., 2008) or convex relaxations such as semidefinite programming (Alessandroni et al., 2023). This ensures our procedure retains the nature of the original combinatorial problem.
>
> ### Performance Guarantees:
> Our approach is guaranteed to match or outperform the heuristics in (Mücke et al., 2025), as our MDP-based method explores a strictly richer decision space. We agree that tighter performance bounds for the full method are an exciting direction for future work.
>
> ### Runtime Considerations and Practical Efficiency
> While our method has exponential worst-case complexity in the number of matrix updates $T$, we mitigate this in practice:
> #### Policy Rollout:
> We limit full tree expansion by switching to a base policy after a small rollout horizon, reducing computational overhead significantly.
>
> #### Impact-Based Index Selection (IMPACT):
> Instead of branching on all matrix entries, we restrict updates to the few entries that directly affect the DR. This reduces the branching factor without noticeable degradation in performance, as shown in our experiments.
>
> #### Efficient Bound Computation:
> Our pruning bounds can be computed in $O(Tn^2)$ time (Appendix C), making the B&B framework practically efficient as a preprocessing step—even for moderately sized QUBO problems.
>
> Thus, while the method remains heuristic, its computational profile is controllable, and it does not become a prohibitive bottleneck in practice.
>
> ### Novelty of Contributions
>
> We respectfully highlight several novel aspects of our work:
> - We are the first to formulate precision reduction for QUBO as a long-sighted Markov Decision Process, enabling more globally informed decisions than greedy heuristics.
> - We introduce a principled Branch-and-Bound algorithm with provable bounds and policy rollout integration.
> - Our approach is general-purpose, improving over (Mücke et al., 2025) while being applicable to arbitrary QUBO instances—unlike many prior methods which are problem-specific or depend on particular constraints.
> - Finally, we demonstrate practical relevance by improving real hardware solver performance (e.g., QA and DA), including power and resource usage on FPGA designs.
>
> We are grateful for the reviewer’s encouraging comments on the clarity of the paper and the relevance of our low-level optimization perspective. We will incorporate additional clarifications on theoretical guarantees in the final version.

---

### Decision · Program_Chairs · 2025-05-01

**Decision:**

Reject

**Comment:**

This paper studies the Quadratic Unconstrained Binary Optimization problem, a fundamental problem for the optimization and ML community. The authors suggest a branch-and-bound procedure to reduce the required bit complexity for that problem. While all reviewers agreed that this is, in principle, an interesting problem to study, there were a couple of concerns: the studied cases of the QUBO seem to be narrow special cases, the empirical validation is too small-scale, and the proposed method could be studied deeper from a theoretical perspective. While some of these issues could be resolved during the rebuttal, still none of the reviewer championed the paper and it seems that a resubmission with the issues being addressed in the first place would be the better thing to recommend here. I encourage the authors to do so!